# The Translation of Nanomedicines in the Contexts of Spinal Cord Injury and Repair

**DOI:** 10.3390/cells13070569

**Published:** 2024-03-24

**Authors:** Wenqian Wang, Joel Yong, Paul Marciano, Ryan O’Hare Doig, Guangzhao Mao, Jillian Clark

**Affiliations:** 1School of Chemical Engineering, University of New South Wales (UNSW), Kensington, NSW 2052, Australia; wenqian.wang@mq.edu.au (W.W.); joel.yong@unsw.edu.au (J.Y.); guangzhao.mao@unsw.edu.au (G.M.); 2Adelaide Medical School, University of Adelaide, Adelaide, SA 5005, Australia; paul.marciano@adelaide.edu.au (P.M.); ryan.doig@sahmri.com (R.O.D.); 3Neil Sachse Centre for Spinal Cord Research, Lifelong Health Theme, South Australian Health and Medical Research Institute, North Terrace, Adelaide, SA 5000, Australia

**Keywords:** nanomedicine, spinal cord injury, nanomaterial, nanoparticle physico-chemistry, nanocarrier drug delivery systems, tissue engineering, evidence translation, neuroprotection, neuro-regeneration, immunomodulation

## Abstract

Purpose of this review: Manipulating or re-engineering the damaged human spinal cord to achieve neuro-recovery is one of the foremost challenges of modern science. Addressing the restricted permission of neural cells and topographically organised neural tissue for self-renewal and spontaneous regeneration, respectively, is not straightforward, as exemplified by rare instances of translational success. This review assembles an understanding of advances in nanomedicine for spinal cord injury (SCI) and related clinical indications of relevance to attempts to design, engineer, and target nanotechnologies to multiple molecular networks. Recent findings: Recent research provides a new understanding of the health benefits and regulatory landscape of nanomedicines based on a background of advances in mRNA-based nanocarrier vaccines and quantum dot-based optical imaging. In relation to spinal cord pathology, the extant literature details promising advances in nanoneuropharmacology and regenerative medicine that inform the present understanding of the nanoparticle (NP) biocompatibility–neurotoxicity relationship. In this review, the conceptual bases of nanotechnology and nanomaterial chemistry covering organic and inorganic particles of sizes generally less than 100 nm in diameter will be addressed. Regarding the centrally active nanotechnologies selected for this review, attention is paid to NP physico-chemistry, functionalisation, delivery, biocompatibility, biodistribution, toxicology, and key molecular targets and biological effects intrinsic to and beyond the spinal cord parenchyma. Summary: The advance of nanotechnologies for the treatment of refractory spinal cord pathologies requires an in-depth understanding of neurobiological and topographical principles and a consideration of additional complexities involving the research’s translational and regulatory landscapes.

## 1. Introduction

Spinal cord injury (SCI) is a complex neuropathological condition that leads to significant morbidity and disability. When the spinal cord is damaged or affected by disease, it creates a hostile inflammatory tissue environment that, together with a developmentally programmed growth-inhibitory environment, is nonconductive to functional repair. The complex interplay between inflammatory and growth-inhibitory mediators poses a significant biological barrier to effective treatment. Similar to the brain, the spinal cord is protected by a microvascular barrier called the blood–spinal cord barrier (B-SCB), which hinders traditional drug delivery [1,2]. Nanomaterials, with their unique physicochemical properties, hold immense clinical potential, as exemplified by recent advances in virology, oncology, orthopaedics, and reconstructive surgery. This scientific review aims to provide a broad overview of the current research on nanomedicine for the treatment of SCI. We will explore how functionalised nanoparticles are revolutionising drug delivery, modulating biological responses, and, ultimately, may be engineered to reconstruct damaged tissue or restore function after injury. Additionally, we will discuss recent advances in the field of combinatorial therapy using stem cell-derived and -engineered nanoparticles and present future directions for research in this exciting and rapidly evolving field.

## 2. Nanomedicine for SCI: A Technological Overview

### 2.1. Nanoparticles

Nanomaterials are materials with at least one dimension in the range of 1–100 nm. Nanoparticles (NPs) are a type of nanomaterials with all three dimensions in the nanometre range (also referred to as zero-dimensional nanomaterials). Their small size and high surface-to-volume ratio enhance their interaction with biological systems, facilitating efficient drug delivery and cellular uptake [3]. The physicochemical characteristics (size, shape, surface chemistry, and biocompatibility) of nanoparticles are crucial considerations for the in vivo stability of nano drug delivery systems to the CNS, as well as various colloidal forces that may impact their stability in biological media [4,5,6,7]. While the question of whether there is a single “one-size-fits-all” NP treatment strategy for SCI remains elusive, recent research demonstrates that tailoring NP properties is effective for SCI treatment. However, this is not well summarised in the specific context of SCI.

### 2.2. Nanoparticle Size

The nanoparticle penetration of the blood–spinal cord barrier (B-SCB) and subsequent parenchymal accumulation is a complex interplay between NP size and delivery methodology, cargo, and /or the target cell/mechanism of interest [8,9,10]. Larger NPs, such as those sized 190 nm and 500 nm, have been shown to induce less cell membrane damage and limit infiltration of pro-inflammatory monocytes into the injury site, improving functional recovery in SCI rodents [11,12]. In contrast, smaller NPs, such as those sized 60 nm, have been found to be effective in repairing spinal cord tissue and reducing inflammatory responses [13]. The demonstration by Kurokawa et al. that the size augmentation of NPs (secondary to their aggregation) can switch their B-SCB transit mechanism from endocytosis to micropinocytosis is of interest [14]. Data on micropinocytosis in CNS cell types are sparse and almost exclusively derived from primary neuronal or glial cell lines [15]. Nanoparticle size also appears to play a direct role in extracellular matrix (ECM) motility, with nanoparticles > 100 nm in diameter being unable to transit normal ECM [16]. Other studies have revealed the importance of the NP size–cytotoxicity relationship [17,18]. Examining the effects of three differently sized silver NPs in MC3T3-E1 and PC12 cell lines, Kim et al. reported that smaller NPs (10 nm versus 50 or 100 nm in diameter) elicit pronounced cytotoxicity, BBB breakdown, neuronal injury, and myelin vesiculation [19]. Accordingly, larger-sized NPs (15–150 nm in diameter) are more efficiently phagocytosed [9]. Thus, a larger particle size allows for more efficient drug loading and reticuloendothelial system clearance.

### 2.3. Nanoparticle Shape

The relationship between the shape of NPs (spherical, cylindrical, conical, tubular, hollow core, and spiral) [20] and therapeutic efficiency is primarily attributed to the interactions between shape and biological systems. This includes the size-dependent ability to traverse biological barriers, intracellular uptake, and biodistribution. Initially, there was interest in the ease of synthesis and agility of spherical nanoparticles in navigating complex biological environments. However, subsequent investigations confirmed the superior biodistribution and enhanced cellular uptake of non-spherical rod- or tube-shaped nanoparticles necessary for optimising therapeutic efficacy [6].

The shape of nanoparticles also affects their ability to traverse the intact blood–brain barrier, which is critical for systemic drug delivery [21]. Additionally, nanoparticle shape can influence the kinetics of drug release and, therefore, impact dose response. Nanoparticles with a larger surface area-to-volume ratio are advantageous, as they allow for controlled and sustained drug release. Therefore, careful consideration of shape and attention to physicochemical properties can optimise drug kinetics and achieve desirable patient outcomes (Table 1) [6,21].

### 2.4. Nanoparticulate Surface Chemistry

In addition to the shape of NPs, surface chemistry can be customised to enhance therapeutic effectiveness. This is because surface modifications impact the interaction of nanoparticles with biological systems, including their ability to cross biological barriers, be taken up by cells, and be distributed within the body. Negatively charged glycosylated proteins are at the surface of almost every cell and extracellular matrix (ECM) [32]. As a result, positively charged (cationic) nanoparticles are more easily taken up by cells compared with neutral or negatively charged nanoparticles, which can enhance drug delivery [6]. Conversely, because of their strong nonspecific interactions with proteins and cells, cationic NPs elicit greater cytotoxicity [6].

Surface modifications also affect molecular pathways, inflammatory responses, cellular adhesion, endocytosis, and the intracellular transport of nanoparticles. For example, the protein corona formed on nanoparticle surfaces can affect opsonisation, cellular uptake, and degradation within endo-lysosomal pathways. Negatively charged polystyrene nanoparticles with sulfone or carboxyl groups demonstrate resistance to surface protein adsorption. Surface modifications can also improve antibiotic delivery and bioavailability [26], a finding which has important implications for the rise of antibacterial resistance. Additionally, the coating layer of nanoparticles can enhance radiation efficacy in radio-diagnostic imaging, but the release of reactive oxygen species by electron-emitting nanoparticles can present biological challenges. Further research will be necessary to fully understand the potential advantages and disadvantages of this approach for clinical use [5,22,26,33,34,35].

The physicochemical and mechanical properties of nanosized drug carriers, such as size, shape, charge, and hydrophilicity, strongly affect their circulation time [1]. Designing the size, charge, and shape of biomaterials appropriately can overcome pharmacodynamic limitations [36]. However, it is important to note that autonomic physiology is significantly altered in individuals with SCI [37,38]. In acute SCI hypotension, bradycardia, peripheral vascular pooling, and peripheral oedema may influence systemic blood circulation time and drug clearance via the macrophage reticuloendothelial system [36,37,38]. The exchange of cerebrospinal fluid (CSF) and interstitial fluid (ISF) through the perivascular glymphatic clearance system is also fundamentally changed in the presence of CNS pathology [35]. Although the physiology of the glymphatic system is not fully understood, there is increasing evidence of its connection to CSF bulk flow, noradrenergic tone, and drug distribution. Delivering drugs to the healthy CNS remains challenging due to the presence of significant biological barriers [39]. Therefore, future drug investigations must consider the physiological consequences of intraspinal pathology.

### 2.5. Nanoparticle Biocompatibility

The physicochemical properties of nanoparticles, such as materials, composition, and functionalisation, play key roles in modulating their biological activities [35]. Included in this are the biocompatibility and uptake efficiency of NPs [33]. Nanoparticles can be composed of various materials, including polymers (poly(lactic-co-glycolic acid) [PLGA], polycaprolactone, etc.), liposomes, metals or ceramics (gold, silver, iron oxide) or biological entities [6]. Nanoparticle composition influences a range of pharmacodynamic parameters, inclusive of aptitude for B-SCB penetration, membrane–receptor interactions, endocytosis, and intracellular trafficking [6]. Nanoparticle moieties most intensely investigated for their potential to modulate CNS pathophysiology include (i) polymer nanoparticles, such as PLGA, polycaprolactone, or chitosan nanoparticles (ChNPs) due to their low CNS toxicity and proven efficacy [6,17,40,41]; (ii) liposomes composed of amphipathic lipids; (iii) inorganic metal or ceramic nanoparticles, such as gold, silver, and iron oxide [6]; and (iv) biological entities (exosomes/extracellular vesicles (Evs)/viral vectors, miRNAs) [18,27,42,43,44,45,46,47,48].

Polymer nanoparticles offer the advantages of their “tuneable” properties, biocompatibility, and potential for sustained drug release. However, the challenge lies in achieving precise targeting and controlled receptor-mediated drug release. Liposomes are advantageous because they can encapsulate both hydrophilic and hydrophobic drugs despite challenges related to in vivo stability and scalability. Inorganic nanoparticles have unique physical properties that make them suitable for neuroimaging and therapeutic applications, but their potential for cytotoxicity and longer-term intracellular accumulation needs careful evaluation. Viral vectors are efficient in delivering transgenes, and MSC-derived exosomes also show promise as a biological and potentially safer cell-free alternative for promoting functional recovery after spinal cord injury. Regarding biological entities, Guo et al. demonstrated the effectiveness of intranasally administered mesenchymal stem cell-derived exosomes (MSC-Exo) loaded with phosphatase and tensin homolog small interfering RNA (ExoPTEN) in attenuating PTEN expression, enhancing axonal growth and neovascularisation, and inhibiting microgliosis in a rat model of SCI [47]. Encouraging results were obtained by transiently expressing viral vectors for the delivery of neuro-regenerative gene therapy [48]. Targeted axoplasmic delivery of therapeutic transgenes to the ventral horn avoided the aberrant synaptic plasticity and cytotoxic immunogenicity observed with permanent gene expression or untargeted delivery. One example is adeno-associated viral vector (AAV) transgene delivery approaches in animal models of acute SCI. Implanted self-assembling nano-peptide scaffolds seeded with bone marrow-derived MSC, and brain-derived neurotrophic factor (BDNF) expressing-AVV transgene elicited significant hind-limb recovery in BDNF-AVV treated animals [28]. Thus, while still in experimental stages, the use of viral vectors for transgene therapy in acute SCI holds promise, although the advantages must be weighed against unresolved translational issues.

An important concept is that nanoscale entities can cross the B-SCB and efficiently deliver drugs to the spinal cord parenchyma [21]. Nanotherapeutics for spinal cord pathology treatment target several mechanistic aspects to attenuate complex pathophysiology and promote neuro-recovery (Table 2). Primary mechanistic targets include (i) inflammatory-immune responses and reactive microgliosis (“immunomodulators”), (ii) cellular apoptosis (“neuroprotectives”), and (iii) axonal regeneration failure (“neuroregeneratives”), or combinations thereof (see Table 2). The closely aligned field of regenerative rehabilitation has been reviewed elsewhere and is beyond the scope of the present discussion.

## 3. Emerging Nanomedicine-Based Strategies for SCI Therapy

### 3.1. Cellular Nanotherapeutics

Cellular nanotherapeutics involve the engineering and targeting of specific multipotential cell types or cell-derived biological entities (EVs, exosomes, microvesicles, miRNA, apoptotic bodies) to modulate a range of histological, anatomical, and behavioural outcomes [18,43,47,52,53,54,55,56,57,58,59,64,65,66,67,68,69,70,71]. Strategies include the delivery of stem/progenitor cells [52,53,57], stem cell-derived EVs or exosomes [18,43,47,58,59,70], and combined cell-based therapies [54,56,64,65,66,67,68,69] encompassing stem cell- or exosome-loaded implantable scaffolds. Apropos of regulatory approval, the NIH https://clinicaltrials.gov website (accessed on 12 January 2024) lists [12] stem cell clinical trial protocols enrolling participants with acute spinal cord injury. A 2022 metanalysis by Shang et al. [72] completed allogeneic or autologous stem cell clinical trials, enrolling 2439 participants with a diagnosis of SCI cited safety concerns [73]. Analysis of adverse events by type (28) and frequency (20%) indicated a propensity for iatrogenic stem cell exposure to elicit or exacerbate neurogenic sequelae [73]. A 2018 meta-analysis by [38] extracted preclinical studies indicated the positive involvement of prevaricating variables in their results. Abbazadeh et al. reported statistically significant effect sizes for stem cell species (neural > iPSC > MSC), administered dose (3 × 10^6^ cells per kg > lower dosages), and injury phase (acute/subacute > chronic) and favourable behavioural outcomes [74]. Regarding principled translation, interspecies differences in genetics, biology, and physiology, as well as clinical heterogeneity and treatment confounders, may each play a role, implying a need to assemble robust research evidence [38].

In relation to biological factors affecting the aforementioned results, stem cell-secreted single membrane EV organelles demonstrate marked heterogeneity in terms of their protein composition, protein enrichment and coding or non-coding RNA content [72,75]. Knowledge gaps pertaining to the “molecular machinery of exosome biogenesis and release” [76] point towards a need to obtain in vivo information relevant to our understanding of endogenous EV/exosome biogenesis, intercellular trafficking, and exosome communication [77]. Introducing additional levels of complexity, the precise mechanism(s) whereby miRNA content is loaded into exosomes remain unresolved [72]. Reviewing knowledge pertaining to exosome-loaded miRNA carriers for SCI, Shen and Cai [2] listed several unresolved issues, and there is clearly more to learn about exosome-miRNA synergism, heterogeneity, especially under pathological conditions [6], pleiotropy, and pharmacology [72,77].

### 3.2. Cell-Free Nanotherapeutics

The term “cell-free” refers to the use of nanomaterials designed to deliver therapeutic agents to the injured spinal cord without the direct involvement of cells. One advantage is that nanomaterials offer several unique properties that facilitate targeted, intraspinal delivery of therapeutic agents, thereby minimising off-target effects [78]. However, cell-free nanotherapeutics present several translational challenges, particularly when implanted in a surgical context, and further research is needed to determine their clinical efficacy [79]. The complexity of spinal cord pathophysiology, as well as clinical heterogeneity to which demographics, genotype–phenotype interactions, treatment-related variables, and the lived experience importantly and diversely contribute, also present a range of methodological challenges requiring careful consideration of the interface between nanotechnology, human biology, research ethics, and the specific needs of individual patients [79].

Table 2 summarises key strategies and mechanistic targets in this field. These include but are not limited to nanoparticle-based drug delivery, in situ nanotherapeutics, and biochemical and physico-mechanical cues. First, nanoparticles can be used to deliver and enhance the biological activities of single-molecule therapeutics targeted to modulate drug-receptor interactions [5,21]. Second, in situ nanotherapeutics involves the use of nanomaterials to deliver therapeutic nucleic acids, such as RNAi, to sites of secondary pathophysiology. The theoretical goal is to target specific molecular pathways signalling ROS-mediated lipid peroxidation, damage-associated molecular pattern-induced inflammasome activation, or autophagy. A third and exciting new development involves the engineering of nanomaterials to recapitulate programmes instructive to the topographic organisation of spinal cord anatomy during embryonic development. Observations of glial mechanosensing in the developing retina and Alzheimer’s disease mutant mice, together with putative roles in glymphatic physiology, indicate that the mechanical properties of nanomaterials might be exploited to mimic physiological signals [80,81]. Nanotherapeutics modulating the stem cell mechanosensing/transduction machinery are currently under development for bone repair, and the question now is whether this new paradigm might translate into neural repair.

### 3.3. Combinatorial Nanotherapeutics

Whereas single delivery systems enable the targeted delivery of a therapeutic agent (e.g., receptor antagonist, agonist, or growth factor [5]), combinatorial approaches allow for the co- or phased delivery of two or more drugs targeted towards discrete mechanisms [53,60,65,82,83,84]. As an example, Braga et al. [54] combined in situ lipocalin 2 (Lcn2) packaging RNA-small interfering RNA (pRNA-RNAi) to generate an immune-compatible niche conducive to iNSC survival [54]. The authors’ demonstration in treated mice (subacute thoracic spinal cord contusion) that Lcn2 pRNA-RNAi enhances oligodendrocyte precursor proliferation and attenuates reactive gliosis confirms proof-of-concept [54]. Recent research in cancer therapeutics offers insight into how these multifunctional, combinatorial nanomedicine approaches might be applied.

## 4. Nanoneuropharmacology

The common delivery routes for nanomedicine targeting spinal cord pathology are intravenous, intrathecal, intranasal, or intraspinal administration. Nanoparticles can be administered intravenously to allow for systemic distribution throughout the body and potential transport across the B-SCB [12,45,49,50,85,86,87,88,89,90,91,92,93,94]. Gao et al. demonstrated that intravenously administered nanoparticles are localised preferentially and dose-dependently at the lesion site in the spinal cord. Additionally, the nanoparticles localised at the lesion site were shown to be retained for an extended period of time (>1 week) [85]. Urdzikova et al. compared the effects of intravenous injection of MSCs, a freshly prepared mononuclear fraction of bone marrow cells or granulocyte colony-stimulating factor-induced bone marrow cell mobilisation in rats with balloon-induced spinal cord compression [90]. Xu et al. reported a delivery system based on nanocapsules (2-methacryloyloxyethyl phosphorylcholine-co-polylactic acid) that allows intravenously injected nerve growth factor (NGF) to enter the CNS. In mice with SCI, the intravenous delivery of NGF promotes neural regeneration, tissue remodelling, and functional recovery [92]. The systemic administration of drugs represents challenges in spinal cord injury, as repeated administration may cause adverse effects on other organs, and achieving a high concentration of the drug at the injury site is difficult. Furthermore, the development of anti-PEG antibodies or acquired hypersensitivity to PEGylated therapeutics in patients is an important safety precaution [95].

Intrathecal administration involves the delivery of nanoparticles directly into the cerebrospinal fluid surrounding the spinal cord, allowing for targeted delivery to the spinal cord parenchyma [5,96,97,98,99,100]. Tukmachev et al. designed a magnetic system in order to accumulate stem cells at a specific intraspinal lesion site after intrathecal administration [101]. They achieved this by loading the stem cells with engineered superparamagnetic iron oxide nanoparticles (SPIONs) that generate sufficient attractive magnetic forces to enable rapid and precise guidance of the SPION-labelled cells to the lesion location. A histological analysis of cell distribution throughout the cerebrospinal fluid system revealed a satisfactory correlation with the calculated distribution of magnetic forces exerted on the transplanted cells. These findings suggest that the proposed non-invasive magnetic system can achieve the focused targeting and fast delivery of stem cells, and NPs can be injected directly into the spinal cord parenchyma at the injury site for the localised delivery of therapeutic agents [24,62,102,103]. Wang et al. explored the use of biodegradable PLGA nanoparticles for the efficient delivery and sustained release of glial cell line-derived neurotrophic factor in treating SCI via intraspinal administration [24]. The study found that these nanoparticles were well absorbed by neurons and glia, effectively preserving neuronal fibres and improving hind-limb locomotor recovery in treated rats, suggesting a potential treatment strategy for SCI [24].

Intranasal administration is a non-invasive olfactory route that delivers nanoparticles to the CNS, including the spinal cord, bypassing the microvascular B-SCB [47,61]. Studies have demonstrated that the intranasal administration of MSC-derived exosomes loaded with PTEN siRNA (ExoPTEN) can migrate to the injured spinal cord, reduce neuroinflammation, and significantly improve functional recovery in rats with complete spinal cord injury, suggesting a potential clinical application [47]. Intramuscular administration delivers nanoparticles to the spinal cord, bypassing the peripheral host defence system and B-SCB. Mao et al. formulated a world-first nanomedicine by combining an adenosine receptor antagonist drug, WGA-HRP, and gold (Au)NP in a tripartite nanoconjugate form to selectively deliver the drug to respiratory motor neurons in the spinal cord and brain, restoring lost respiratory functions in a rat model of cervical spinal cord injury [25,104]. This nanomedicine design is based on the knowledge of the WGA-HRP tracing of the crossed-phrenic phenomenon [105] while taking advantage of gold nano-chemistry [106]. Specifically, the nanotherapeutic design consists of a transsynaptic tracer, WGA-HRP, chemically conjugated to an AuNP, which in turn is chemically conjugated to a pro-drug, pro-theophylline, or pro-DPCPX via a biodegradable bond. Injecting WGA-HRP into the diaphragm muscle results in its uptake by the terminals of phrenic axons and retrograde transport to phrenic motor neurons.

Distinct delivery routes present different advantages and challenges in intraspinal pathology. Intravenous administration allows for systemic distribution but may have limited penetration of the B-SCB, while intrathecal and intraspinal administration allow for more targeted delivery but are more invasive. Intranasal and intramuscular administration offer ‘bypassing routes’, but there may be limitations in terms of drug delivery efficiency. The choice of delivery route depends on the specific characteristics of the nanomedicine and the therapeutic goals of the treatment.

## 5. Challenges and Future Directions

### 5.1. Neurotoxicity

While nanomaterials have shown promise in treating SCI, it is also important to consider their biocompatibility and potential side effects. It is widely agreed in the scientific community that the biocompatibility and degradation of nanoparticle-based implants or scaffolds can be improved. Poor biocompatibility can lead to adverse reactions, while degradation issues can affect the long-term effectiveness and safety of the treatment [61]. In a study by Raspa et al., the biocompatibility of two types of coaxially electrospun microchannels was tested using in vitro and in vivo assays [23]. The results showed that the first type, consisting of a core of poly (ε-caprolactone) and PLGA (PCL–PLGA) and a shell of an emulsion of PCL–PLGA and a functionalised self-assembling peptide Ac-FAQ, had better cell viability and tissue response compared with the second type, which had a core of Ac-FAQ and a shell of PCL–PLGA. The authors suggested that the emulsification of the outer shell improved the biocompatibility of the scaffolds by enhancing the interaction between the self-assembling peptides and the cells.

Nanoparticles used for the treatment of SCI may carry neurotoxic risks, potentially triggering neuroinflammation, neurodegeneration, and other neurotoxic effects that can have adverse consequences such as altered neuronal structure or activity, glial activation, and glial–neuronal interactions, with the potential for reversible or permanent effects on the CNS. The size and particulate chemistry of nanoparticles play important roles in determining their neurotoxicity, and further in vivo studies are required to fully understand their impact. Additionally, the potential neurotoxic effects of nanomaterials must be carefully evaluated as they are increasingly being used in biomedicine for various purposes, including drug delivery and bioimaging [107].

The neurotoxicity or neuroprotection induced by nanoparticles depends on their exposure and usage. For example, systemic exposure to engineered nanoparticles made of metals or silica dust worsened the outcome of SCI in animal models. However, when drugs were conjugated to titanium nanoparticles or encapsulated in liposomes, their neuroprotective effectiveness following SCI was enhanced [108]. Yuan et al. evaluated the in vivo neurotoxicity of Tween 80-modified chitosan nanoparticles after intravenous injection in rats [109]. The results showed a dose-dependent accumulation of the nanoparticles in the brain, neuronal apoptosis, inflammatory response, increased oxidative stress, and weight loss. In a mouse model, these same nanoparticles induced an inflammatory response in the frontal cortex, while cerebellar glial fibrillary acidic protein expression was decreased. Dendrimers have been found to induce several neurotoxicological responses [29]. The effects of polyamidoamine dendrimers on a 3D neurosphere system using human neural progenitor cells were evaluated. The results showed that higher concentrations of dendrimers inhibit cell proliferation and migration [110], while surface functionalisation with polyethylene glycol or folate reduces their neurotoxicity [111]. Inorganic nanoparticles such as gold, silver, iron oxide, titanium oxide, and silica have been shown to translocate into the brain after entering the body. Furthermore, due to their limited excretion, they accumulate in the brain, causing damage to neuronal cells and functional impairments [51]. Studies have reported alterations in synaptic transmissions and nerve conduction, leading to neuroinflammation, apoptosis, and immune cell infiltration due to iron oxide nanoparticles [112]. Studies have also demonstrated that the intranasal delivery of silica nanoparticles leads to the accumulation of nanoparticles in the brain, resulting in cognitive dysfunction and impairment, synaptic changes, and pathologies similar to neurodegeneration [113].

### 5.2. Inflammation and Immunity

Inflammatory responses following SCI can cause extensive tissue damage that impairs function. Nanoparticles have the ability to modulate these responses, but it is important to ensure that the modulation is appropriate and not excessive, as it could potentially exacerbate inflammation and tissue damage [88]. The size, shape and surface properties of NPs greatly affect the type and magnitude of immune response to nanotherapeutics. For instance, in a mouse ovalbumin model, spherical nanoparticles induced a T helper 1 cell-biased (cell-mediated) response, micrometre-length rods induced a T helper 2 cell-biased (humoral) response, and spherical NPs induced a stronger overall immune response [30]. Huo et al. demonstrated that the administration of drug-free liposomes induced neuropathologic changes in rats [31]. This is supported by the fact that nanoparticles, due to their small sizes (10–100 nm), exert higher inflammatory potential compared to the larger particles of the same materials when exposed to cells or tissues [31,108].

### 5.3. Oxidative DNA Damage

Selective delivery of antioxidant nanomaterials or redox NP has shown great promise for the treatment of brain-related neurodegenerative diseases, such as Alzheimer’s disease and Parkinson’s disease [114,115,116]. However, whilst ROS mitigation strategies by functional NPs are important considerations for SCI, there is limited literature. Previously, cerium oxide (CO) and manganese dioxide (MnO_2_) NPs have shown promise in reducing reactive oxygen species (ROS) and nitric oxide species (NOS) in SCI models [117,118].

Cerium oxide NPs internalise cells through endocytic pathways and, following intraspinal delivery, regulate ROS and iNOS, pro-inflammatory cytokines, apoptosis, inflammation, and regeneration, leading to reduced lesion areas and improved neurological function following SCI [117]. The fabrication and implantation of hydrogels dotted with MnO_2_ have demonstrated a synergistic effect on the survival, integration, and neuronal differentiation of encapsulated MSCs following spinal cord transection. MnO_2_ NP-dotted hydrogels can regulate the ROS microenvironment of the injured spinal cord by effectively reducing lipid peroxidation by-products (e.g., 4-hydroxynonenal) and oxidative DNA damage. By alleviating the oxidative microenvironment, the implantation of highly viable MSCs in the multifunctional hydrogel resulted in nervous tissue preservation and regrowth [118]. Infiltrating Schwann cells of peripheral origin have been shown to promote neuro-recovery after SCI via mechanisms that involve the mitochondrial actions of their secreted exosome products. Xu and colleagues (2023) reported that cultured Schwann cell-derived exosomes promote mitophagy via an AMP-dependent mechanism in oxygen–glucose-deprived rat PC12 cells [119]. In contusion models of SCI, Schwann cell-derived exosomes attenuate ROS production and induce autophagy via the EGFR/Akt-TOR signalling pathway [119,120].

Conditional AVV DNA repair enzyme expression experiments in mice revealed the association between DNA damage accumulation and the initiation and progression of age-related neurodegenerative pathologies [121]. Subsequent investigations in cancer cell lines confirmed the regulatory role of oxidative DNA damage in cellular apoptosis mediated via programmed cell death-ligand1 [122]. Importantly, the DNA damage response system exerts its activities via multiple base lesion repair pathways. An in-depth discussion of the mechanistic bases of this topic is beyond the scope of this review, but it is important to note that clinical findings in SCI are preliminary [123]. Future investigations may help us understand when, where, and in which etiopathologies redox nanotherapeutics might most effectively be applied.

## 6. Future Perspectives of Nanomedicine for Spinal Cord Injury Repair

A better understanding of the organising principles of intraspinal cell biology and behaviour and CNS microanatomy is opportune for the design of novel and potentially disruptive nanomedicines [124]. The major challenges are, first, to enable the preclinical-clinical trial translational pathway, which may involve the partnering of agnostic high through-put discovery approaches with hypothesis-driven research. Secondly, it will be necessary to resolve the disappointing regulatory approval rate [124]. In this regard, refining the organ-, tissue-, site- and cell-specific biodistribution of nanocarriers and their payload, optimising dose efficacy, and reassessing clinical trial design will be key [125]. Thirdly, the incidence of new index cases of SCI meets the consensus definition of a rare disease [126]. Extrapolating from the rare disease literature, investment in clinical and translational infrastructure, together with financial incentives, such as tax credits, fee exemptions and licencing, will be necessary in order to revert a long and storied history of disappointing clinical trial outcomes [125,126]. Finally, it would be ideal to realise health equity for the cohorts presenting each year and people living with SCI. We would judiciously suggest the benefit of convening an International SCI Nanomedicine Consortium that is positioned to resolve the immediate challenges.

## 7. Conclusions

The intersection of nanotechnology and neuroscience in the field of nanomedicine holds great promise for the treatment of spinal cord injury. However, the principled translation of neuronanomedicines for the treatment of intraspinal pathology requires rigorous preclinical evaluation in animal models, the identification of relevant biomarkers to assess safety and efficacy, and a thorough understanding of the regulatory landscape to ensure the safe translation of nanotherapeutics from the laboratory to the clinic. While challenges exist, the potential of neuronanomedicine to revolutionise the treatment of spinal cord pathology is significant and presents positive opportunities for the future.

Graphical Abstract: Conceptual basis of nanotherapeutic engineering for the treatment of spinal cord injury (SCI): (a) Conventional drugs conjugated with; (b) Cell-derived or cell-free nanomaterials or nanoparticles; (c) Class: immunomodulator, neuroprotective, or neuro-regenerative, administered alone or in co- or phased protocols to; (d) Cellular targets: resident cells (neurons, interneurons, microglia, infiltrating leukocytes, pericytes*, vascular endothelial cells*, and Structural targets: microarchitecture, fine tissue architecture, extracellular matrix components, vessels* *Vascular targets not shown.

Schematic 1: Conceptual bases of (A) cell-derived biopharmacy and cell-free nanoconjugates; (B) pharmacokinetics; (i) cell-free NP-i.t.-csf-; (ii) cell-derived NP-blood-borne-BBB/ B-SCB, (iii) cell-free NP intraspinal injection, (iv) cell-derived-intraspinal scaffold; (v): cell-free- i.c.v.- apertures of Luschka or Magendie-subarachnoid space-csf-; (vi): cell free-intranasal-cribriform plate-BBB-csf-, (vii) cell free-i.m. -NMJ-synaptic cleft-axoplasmic-spinal neuron-brain stem-; (viii) *cell free-i.v.- B-SCB, BBB- apertures of Luschka or Magendie- c.s.f.; (C) cellular targets; (i) neurons; (ii) interneurons; (iii) microglia; (iv) infiltrating leukocytes, (v) migrating Schwann cells; (vi) pericytes/ vascular endothelial cells; (D) cellular internalisation; (i) endocytosis, (ii) (micro)-pinocytosis; (E) clearance: (i) macrophagic reticuloendothelial system. *Of relevance to BBB, B-SCB compromise or altered B-SCB permeability.

## Figures and Tables

**Table 1 cells-13-00569-t001:** Nanoparticle physicochemical–biological advantage relationship.

Kinetics	Biological Advantage	Physicochemical Property (nm)	References
Drug Delivery	BSCB transit; parenchymal accumulation	Size (20–60 nm); Shape (non-spherical; rod/tube)	[8,14]
	ECM motility		[6,21]
	Targeted parenchymal delivery	Size (<100 nm)	[16]
	ECM remodelling; axonal regeneration	Surface modifications (dual action hydrogels)	[3,6,16,22]
Drug Loading	Drug reservoir	Size (<100 nm)	[1,4]
	Parenchymal concentration; penetration	>surface area-to-volume ratio (nanofibres; hydrophobic drugs)	[6,21,23]
Drug Release	Controlled, sustained release kinetics	>surface area-to-volume ratio	[24,25]
Cellular Uptake	Efficient cellular adhesion; endocytosis; trafficking	Surface modifications	[6,21]
	Protein corona/low surface adsorption	Surface negative charge	[3,8,14,26]
Biodistribution	Cellar uptake efficacy; superior biodistribution	Shape (non-spherical; rod/ tube)	[4,5,6]
Biocompatibility	Avoid premature biological exposure	Surface modifications (outer shell; encapsulation)	[23]
Toxicity	Low CNS toxicity/inflammatory response	Composition (PLGA, ChNPs, lipids, biological entities); Size (50–100 nm)	[6,17,18,27,28,29,30,31]
Clearance	Phagocytosis	Size (15–150 nm)	[1]

**Table 2 cells-13-00569-t002:** Nanomedicine translation: biological, mechanistic, and therapeutic targets.

Biological Target	Mechanistic Target	Therapeutic Agent	References
Neuroprotection	Oxidative stress; Nitrosylation	Antioxidant-loaded polymers, liposomes, micelles, dendrimers	[49,50,51]
	Apoptosis	Stem cells, exosomes, EVs, MiRNA, apoptotic bodies, biodegradable PLGA/ GDNF, Therapeutic RNAi, TiO2, MSC-BDNF-AVV	[1,17,27,47,52,53,54,55,56,57,58,59,60]
Immunomodulation	Inflammation	MR-active SPIO@Chitosan-GL/MSC spheroids; Exo PTEN; AU-proDPCPX	[6,25,47,61]
Neurodegeneration	Apoptosis	Biodegradable PLGA/BDNF, PLGA/GDNF	[28,62]
Neuro-regeneration	Reactive gliosis	Tuned collagen, fibrin hydrogel/physiological signals, Lcn2 pRNA-RNAi	[54]
	Neurite growth inhibition	Imidazole-poly(organophosphazenes) (I-5)/arylsulfatase B (ASRB) hydrogel	[63]
	Axonal regeneration failure	Stem cells, exosome-loaded scaffolds, viral vectors	[48,54,56,64]
Neuroplasticity	Aberrant synaptogenesis	Transiently expressing BDNF-AVV, self-assembling amphiphile	[48]

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
