# Peer review of "The Translation of Nanomedicines in the Contexts of Spinal Cord Injury and Repair"

_cells, 2024, doi:10.3390/cells13070569_

Round 1
Reviewer 1 Report
Comments and Suggestions for Authors
In the present review, the main conceptual bases of nanomaterials chemistry and nanotechnology focused on Spinal Cord Injury repair have been addressed. The evaluation and use of nanomedicines are recognized among scientists, pharmacists and regulatory bodies as complex, but smart and of great importance, so it is a strength of the present work and in my opinion, it should be published.
All sections of the article cover and review in a succinct but thorough way the different characteristics that make nanoparticles such a promising tool in medicine in general and in spinal cord injury repair in particular.
No ethical concerns.
SUGGESTED CHANGES
- Page 6, Line 199, “3. Emerging Nanomedicine-Based strategies for SCI therapy”: In order to maintain consistency with the previous headings 1 and 2, it would be better to change 3.0. to 3.
- Add a section on "Future perspectives of nanomedicine for spinal cord injury repair".
Author Response
The Authors extend their gratitude to Reviewer 1# for their valuable time, and comments. Refutes are detailed below for the Reviewer’s consideration.
- Page 6, Line 199, “3. Emerging Nanomedicine-Based strategies for SCI therapy”: In order to maintain consistency with the previous headings 1 and 2, it would be better to change 3.0. to 3.
- Response: We apologise for this oversight. This inconsistency is corrected in the revised manuscript.
- Add a section on "Future perspectives of nanomedicine for spinal cord injury repair".
- Response: Thank you for making this suggestion. The Authors refer the Reviewer to the 5.0 Subheading entitled " Challenges and Future Directions" and the added subheading 5.3 " Oxidative DNA damage" Please also refer to 6. below
6. Future perspectives of nanomedicine for spinal cord injury repair
A better understanding of the organising principles of intraspinal cell biology and behaviour and CNS microanatomy is opportune for the design of novel and potentially disruptive nanomedicines 125. The major challenges are first, to enable the preclinical-clinical trial translational pathway, which may involve the partnering of agnostic high through-put discovery approaches with hypothesis-driven research. Secondly, it will be necessary to resolve the disappointing regulatory approval rate 125. In this regard, refining the organ-, tissue-, site- and cell-specific biodistribution of nanocarriers and their payload, optimising dose efficacy, and reassessing clinical trial design will be key 126. Thirdly, the incidence of new index cases of SCI meets the consensus definition of a rare disease 127. Extrapolating from the rare disease literature, investment in clinical and translational infrastructure, together with financial incentives, such as tax credits, fee exemptions and licencing, will be necessary in order to revert a long and storied history of disappointing clinical trial outcomes126, 127. Finally, it would be ideal to realise health equity for the cohorts presenting each year and people living with SCI. We would judiciously suggest the benefit of convening an International SCI Nanomedicine Consortium that is positioned to resolve the immediate challenges.
References:
Russ, D.E., Cross, R.B.P., Li, L. et al. A harmonized atlas of mouse spinal cord cell types and their spatial organization. Nat Commun 12, 5722 (2021). https://doi.org/10.1038/s41467-021-25125-1
Halapi E, Hakonarson H. Advances in the development of genetic markers for the diagnosis of disease and drug response. Expert Rev Mol Diagn. 2002 Sep;2(5):411-21. doi: 10.1586/14737159.2.5.411. PMID: 12271813.
Cremers S, Aronson JK. Drugs for rare disorders. Br J Clin Pharmacol. 2017 Aug;83(8):1607-1613. doi: 10.1111/bcp.13331. Epub 2017 Jun 27. PMID: 28653488; PMCID: PMC5510061.
Reviewer 2 Report
Comments and Suggestions for Authors
The article entitled The Translation of Nanomedicines in the Contexts of Spinal Cord Injury and Repair has taken into consideration one of the most ambitious challenges of neuron system repair and therefore its correction. For many decades the recovery of the CNS has been the physician's dream however not so much has been achieved in this field - therefore the review is highly demanded. The article decides to discuss the pharmacology bio-pharmacy, and pharmacokinetics and connect them with nano-chemistry. The biological/biochemical part was also focused. The review is very well written without unnecessary digression and due to that is readable. In my opinion can be a valuable position for further young scientists in the field. During the review, I found some points that should be mentioned: the induction of not only ROS but also NOS, the role of oxidative stress in the context of Schwann cells should be discussed as well as the fat acid degradation product. The induction and role of DNA damage in the context of neurodegenerative disorder should be mentioned (a special defects in DDR). Additionally, it would be nice if the authors put some graphics/figures to illustrate the discussed problem. I think that the transport of nanoparticles through BBB should be shown by the scheme.
In conclusion, I recommend the article for publication as valuable and inspiring for a broad audience.
Author Response
The Authors extend their appreciation to Reviewer 2# for sharing their helpful, and constructive comments and suggestions, and their recognition of the significance and merit of nanomedicines for the treatment of spinal cord injury and spinal cord repair. We also thank them for their valuable time. Our responses are detailed below.
-Reviewer:” Schwann cells should be discussed…”
-Response: Thank you for bringing this to our attention. A discussion of PNS cell biology is not within scope. However, we appreciate the roles that the exosome products of cultured and migrating Schwann cells have in the CNS. The manuscript has been revised accordingly (please refer to Section 5.3) and the text/ references shown below.
“Infiltrating Schwann cells of peripheral origin have been shown to promote neurorecovery after SCI via mechanisms that may or may not involve the mitochondrial actions of their secreted exosome products. Xu and colleagues (2023) reported that cultured Schwann cell-derived exosomes promote mitophagy via an AMP-dependent mechanism in oxygen-glucose deprived rat PC12 cells. In contusion models of SCI, Schwann cell-derived exosomes attenuate ROS production (Xu et al 2023) and induce autophagy via the EGFR/Akt-TOR signalling pathway (Pan et al 2022).”
References:
Xu B, Zhou Z, Fang J, Wang J, Tao K, Liu J, Liu S. Exosomes derived from Schwann cells alleviate mitochondrial dysfunction and necroptosis after spinal cord injury via AMPK signalling pathway-mediated mitophagy. Free Radic Biol Med. 2023 Nov 1;208:319-333. doi: 10.1016/j.freeradbiomed.2023.08.026. Epub 2023 Aug 26. PMID: 37640169.
Pan D, Zhu S, Zhang W, Wei Z, Yang F, Guo Z, Ning G, Feng S. Autophagy induced by Schwann cell-derived exosomes promotes recovery after spinal cord injury in rats. Biotechnol Lett. 2022 Jan;44(1):129-142. doi: 10.1007/s10529-021-03198-8. Epub 2021 Nov 5. PMID: 34738222; PMCID: PMC8854309.
2022
-Reviewer: Some points that should be mentioned: “the induction of not only ROS but also NOS, the role of oxidative stress in the context of Schwann cells should be discussed as well as the fat acid degradation product.”
-Response: The Authors thank the Reviewer. This revision reflects knowledge related to NOS/ ROS inclusive of the two in vivo papers that, to our knowledge, appropriately assess redox in traumatic SCI models.
“Selective delivery of antioxidant nanomaterials or redox NPs has shown great promise for the treatment of brain-related neurodegenerative diseases, such as Alzheimer’s Disease and Parkinson’s disease (Sadowska-Bartosz & Bartosz, 2018; Yashitomi & Nagasaki, 2011 Gilmore et al., 2007). However, whilst ROS mitigation strategies by functional NPs are important considerations for SCI treatment strategies, there is a limited literature. Previously, Cerium oxide (CO) and Manganese dioxide (MnO2) NPs, have shown promise in reducing reactive oxygen species (ROS) and nitric oxide species (NOS) in SCI models (Kim et al., 2017; Li et al., 2019). Cerium oxide NP internalize cells through endocytic pathways, and following intraspinal delivery, regulate ROS and iNOS, proinflammatory cytokines, apoptosis, inflammation and regeneration, leading to reduced lesion areas and improved neurological function following SCI (Kim et al., 2017). Fabrication and implantation of hydrogels dotted with MnO2, have demonstrated a synergistic effect on the survival, integration, and neuronal differentiation of encapsulated MSCs following spinal cord transection. MnO2 NP-dotted hydrogels can regulate the ROS microenvironment of the injured spinal cord by effectively reducing lipid peroxidation by-products (e.g. 4-hydroxynonenal) and oxidative DNA damage. By alleviating the oxidative microenvironment, the implantation of highly viable MSCs in the multifunctional hydrogel resulted in nervous tissue preservation and regrowth (Li et al, 2019).
References:
-Reviewer: "The induction and role of DNA damage in the context of neurodegenerative disorder should be mentioned (a special defect in DDR)."
- Response: Thank you for drawing this emerging and interesting concept to our attention. The manuscript has been revised with the caveat that the DDR data pertaining to traumatic SCI are of a preliminary nature. Please refer to 5.3 in this revision and the paragraph reproduced below.
“Conditional AVV DNA repair enzyme expression experiments in mice revealed the association between DNA damage accumulation and the initiation and progression of age-related neurodegenerative pathologies 122. Subsequent investigations in cancer cell lines confirmed the regulatory role of oxidative DNA damage in cellular apoptosis mediated via programmed cell death-ligand1 123. Importantly, the DNA damage response system exert its activities via multiple base lesion repair pathways. An in-depth discussion of the mechanistic bases of this topic is beyond the scope of this review, but it is important to note that clinical findings in SCI are preliminary. 124 Future investigations may help us to understand when, where and in which etiopathologies redox nanotherapeutics might most effectively be applied."
.”
References:
Martin LJ, Wong M. Enforced DNA repair enzymes rescue neurons from apoptosis induced by target deprivation and axotomy in mouse models of neurodegeneration. Mech Ageing Dev. 2017 Jan;161(Pt A):149-162. doi: 10.1016/j.mad.2016.06.011. Epub 2016 Jun 27. PMID: 27364693; PMCID: PMC5192008.
Permata TBM, Hagiwara Y, Sato H, Yasuhara T, Oike T, Gondhowiardjo S, Held KD, Nakano T, Shibata A. Base excision repair regulates PD-L1 expression in cancer cells. Oncogene. 2019 Jun;38(23):4452-4466. doi: 10.1038/s41388-019-0733-6. Epub 2019 Feb 12. PMID: 30755733.
Scheijen EEM, Hendrix S, Wilson DM 3rd. Oxidative DNA Damage in the Pathophysiology of Spinal Cord Injury: Seems Obvious, but Where Is the Evidence? Antioxidants (Basel). 2022 Aug 31;11(9):1728. doi: 10.3390/antiox11091728. PMID: 36139802; PMCID: PMC9495924.
Reviewer: "Additionally, it would be nice if the authors put some graphics/figures to illustrate the discussed problem. I think that the transport of nanoparticles through BBB should be shown by the scheme."
Response: Thank you for making this helpful suggestion. Please refer to the figure caption detailed the revised text and replicated below. The response site will not accept a jpg, and therefore it is not possible to upload the schematic per se.
Schematic 1: Conceptual bases of: A. Cell-derived biopharmacy and cell free nanoconjugates; B. Pharmacokinetics; (i) cell free NP-i.t.-csf-; (ii) cell-derived NP-blood-borne-BBB/ B-SCB, (iii) cell-free NP intraspinal injection, (iv) cell-derived-intraspinal scaffold; (v): cell free- i.c.v. -csf-; (vi): cell free-intranasal-cribriform plate-BBB-csf-, (vii) cell free-i.m. -NMJ-synaptic cleft-axoplasmic-spinal neuron-brain stem-; viii) cell free-i.v.-portal of Luschka- c.s.f.; C) Cellular targets; (i) neurons; (ii) interneurons; (iii) microglia; (iv) infiltrating leukocytes, (v) migrating Schwann cells; (vi) pericytes/ vascular endothelial cells; D) Cellular internalisation; (i) endocytosis, (ii) (micro)-pinocytosis; E) Clearance: (i) macrophagic reticuloendothelial system.